# Formulating Equations for Estimating Forest Stand Carbon Stock for Various Tree Species Groups in Northern Thailand

**Khwanchai Duangsathaporn \*, Narapong Sangram \*, Yenemurwon Omule, Patsi Prasomsin, Kritsadapan Palakit and Pichit Lumyai**

Department of Forest Management, Faculty of Forestry, Kasetsart University, Bangkok 10900, Thailand; ayomule@gmail.com (Y.O.); fforpsp@ku.ac.th (P.P.); fforkpp@ku.ac.th (K.P.); fforpcl@ku.ac.th (P.L.)

\* Correspondence: fforkcd@ku.ac.th (K.D.); narapong.sa@ku.th (N.S.)

**Abstract:** Through this study, we established equations for estimating the standing tree carbon stock based on 24 tree species in multiple size classes in a case study at the Ngao Demonstration Forest (NDF) in northern Thailand. Four hundred thirty-nine wood samples from trees in mixed deciduous forest (MDF), dry dipterocarp forest (DDF), and dry evergreen forest (DEF) were collected using non-destructive methods to estimate aboveground carbon equations through statistical regression. The equations were established based on four criteria: (1) the coefficient of determination ($R^2$), (2) standard error of estimate (SE), (3) F-value, and (4) significant value (*p*-value, $\alpha \leq 0.05$). The aboveground carbon stock (C) equations for standing trees in the MDF was C = $0.0199DBH^{2.1887}H^{0.5825}$, for DDF was C = $0.0145DBH^{2.1435}H^{0.748}$, for DEF was C = $0.0167DBH^{2.1423}H^{0.7070}$, and the general equation for all species/wood density groups was C = $0.017543DBH^{2.1625}H^{0.6614}$, where DBH is tree diameter at breast height, and H is tree total height. The aboveground carbon stock in the DDF, MDF, and DEF was 142, 53.02, and 12 tons/ha, respectively, and the estimated aboveground carbon stock in the Mae Huad sector at the NDF was 61 tons/ha.

**Keywords:** carbon stock; standing-tree carbon equation; Ngao Demonstration Forest

## 1. Introduction

Trees can potentially trap atmospheric carbon through the photosynthesis process [1], which involves the conversion of carbon from carbon dioxide ($CO_2$) to carbohydrates, glucose, and starch that are stored in the leaves, stems, branches, and roots, and contribute to a plant's growth [2]. As such, plants store carbon as living biomass, which becomes a part of the food chain and enters the soil as soil carbon [3]. It is estimated that forests contain 77% of the carbon stored in land vegetation, out of which approximately 60% of carbon is stored in tropical forests, 17% in temperate forests, and 23% in boreal forests [4].

Normally, the carbon stored in trees is estimated as the product of the volume of biomass and the carbon fraction (generally assumed as 0.47 [5]) based on field data collection methods and estimations of different complexity levels [6–8]. Tree biomass can be estimated using either direct or indirect methods. The direct method involves the felling of trees and weighing various tree components [9], while the indirect method involves the use of allometric equations for estimating the tree sample biomass [10].

The biomass or tree volume equations to estimate the tree carbon storage specific to Thailand are inaccurate as the commonly used allometric equations are biased (i.e., they tend to over or under-estimate the tree volume) [11]. Additionally, the existing equations do not cover the major tree species frequently found in forests, such as *Tectona grandis*, teak [12], or various dipterocarp species [13]. This is primarily due to the fact that the estimations are based on equations constructed using the destructive sampling of a relatively small number of trees. Some volume equations use only the diameter at breast height (DBH) as the independent variable and do not include tree height [14,15]. Moreover, some

equations were constructed only to estimate the traded logged volume and did not include the smaller trees [15]. Therefore, a novel approach that addresses these weaknesses is proposed to estimate the standing tree carbon content as a function of tree attributes in a natural forest with different sample tree size classes. This approach would also avoid the felling of trees and would use combustion methods to estimate the real carbon fraction.

This study aims to formulate the standing tree carbon equations to estimate the carbon stocks in three forest types: a mixed deciduous forest (MDF), a dry dipterocarp forest (DDF), and a dry evergreen forest (DEF) at the Mae Huad sector, Ngao Demonstration Forest (NDF) in northern Thailand. The Mae Huad sector has a vast forest cover in the NDF, with several tree species, and is one of the five most important biosphere reserve areas in Thailand [16]. The equations determined in this research to estimate the carbon stock were constructed using specific carbon fractions of tree species without the need to calculate the biomass to estimate the tree carbon stock. The non-destructive method used to establish the equations for many tree species sampled from the MDF, DDF, and DEF can also be used to estimate the carbon stored in other sites in Thailand.

## 2. Materials and Methods

The methodology consists of 3 steps, i.e., forest inventory and sample collection, sample preparation and carbon fraction analysis, data analysis including carbon storage in the wood sample, the calculation of standing tree carbon stock, constructing the standing tree carbon equation, and estimating stand carbon stock. These steps are described in detail in Figure 1.

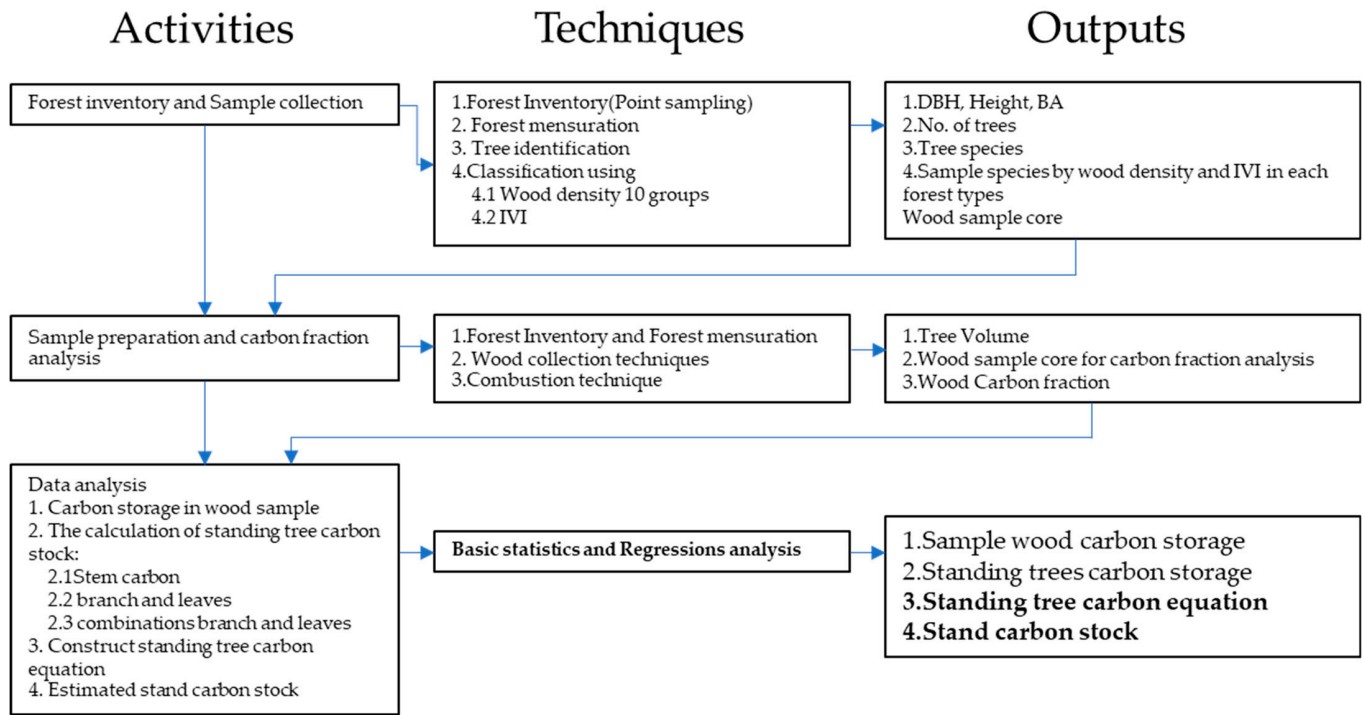

**Figure 1.** Schematic diagram of the study methodology.

### 2.1. Study Area

The Mae Huad Sector was chosen as the study site (Figure 2) and is one of the four designated sectors in the NDF. The NDF consists of four sectors, Mae Heang, Mae Huat, Mae Ngao, and Mae Teeb, and covers an area of approximately 43,431.75 hectares, including several forest types. It is located in the north-western part of the Lampang Province in northern Thailand between 18°30′ and 18°54′ north latitude and 99°50′ and 100° east. The NDF was established in 1961 and is the only demonstration forest in Thailand; and has a long history of functioning as a base for the introduction, testing, and adaption of

new forest management techniques [15]. Most of the land in the Mae Huad sector is under
forest cover, i.e., 38,557.50 hectares or 84.246% of the total area. Most of the tree cover is
part of the Ngao Demonstration Forest, while a total of 6526.80 hectares is classified as agri-
cultural land, or 14.261%, and is located in the national reserve forest by law. The forest
area of the Mae Huad sector includes mixed deciduous forest, MDF (67.26%), dry diptero-
carp forest, DDF (20.87%), dry evergreen forest, DEF (3.59%), and teak plantation (8.27%).
The topography of the Mae Huad sector consists of hill ridges. The elevations vary from
200 m to 1400 m above mean sea level. The geography of this study area showed that re-
cent alluvial terraces are characterized by alluvial deposits that were transported through
the river and streams. Soil textures in this area vary from sand to clay. The climate of the
year is divided into 3 seasons, i.e., the hot season from February to May, the rally season
from June to September, and the cool season from October to January. The average annual
rainfall is about 1117.3 cm.

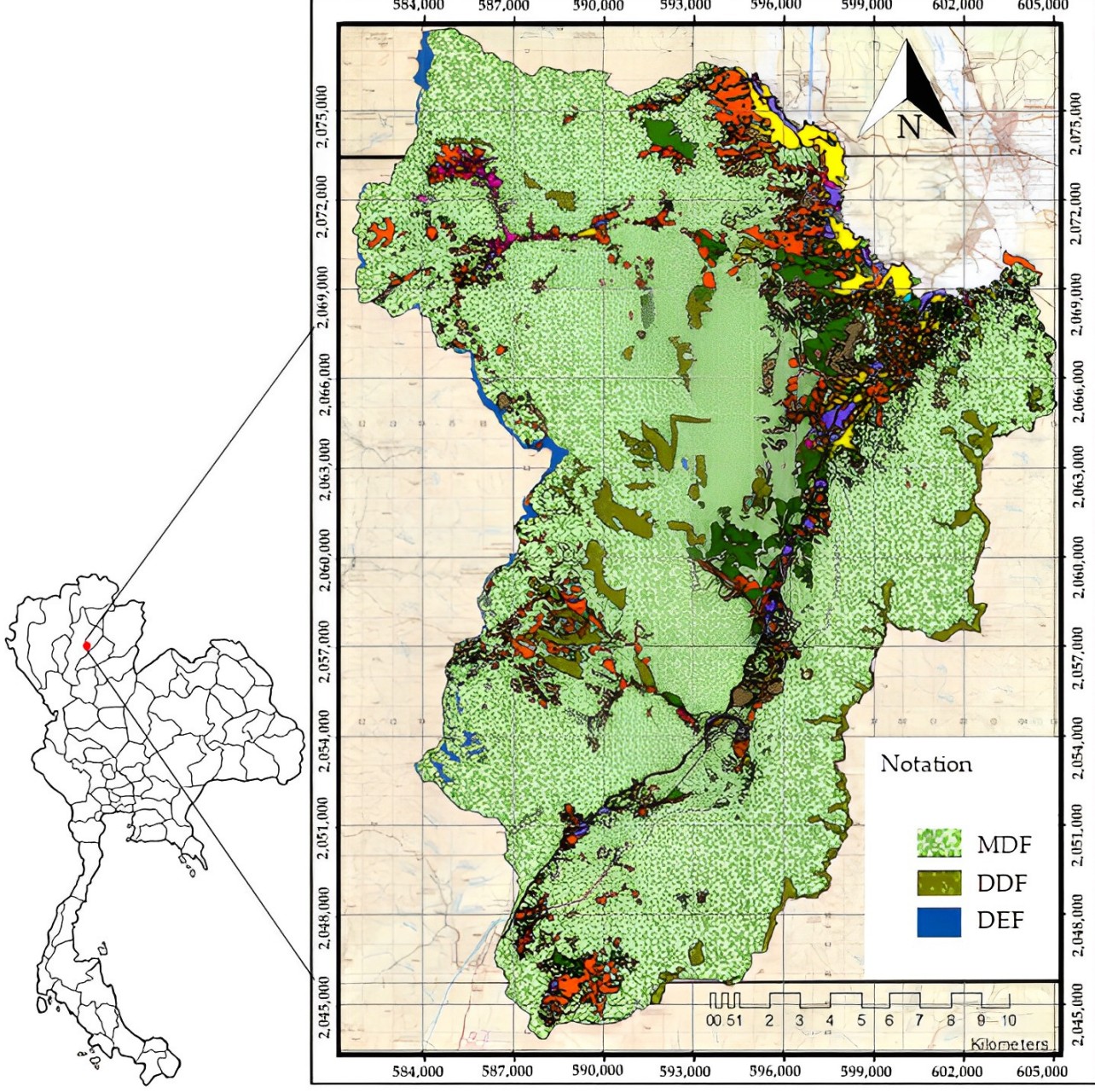

**Figure 2.** The study area of the Mae Huad sector, NDF, in northern Thailand.

### 2.2. Forest Inventory and Sample Collection

Trees from plotless inventory data are used for the collection of tree samples, wood samples, and calculated stand carbon stock. The distribution of trees in the Mae Huad sector NDF, Northern Thailand, was determined using stratified sampling [17,18] and a uniform fixed grid of $3 \times 3$ km systematic arrangement that covered the whole of the Mae Huad sector. The point sampling technique [19–21] was used to collect tree data which included diameter at breast height (DBH), total height (H), tree species, and forest type at each sample point. This grid and point sampling were part of the APFNET project [15]. The point sampling data were used to calculate the importance value index (IVI), which is the quantitative value for measured dominance of tree species [22] that was used to select the sample trees by diameter classes. The suitable sample size (i.e., the number of sampling points) was calculated using Equation (1) [23]

$$n = \frac{t^2 (cv)^2}{AE^2} \tag{1}$$

where, n is the target number of sample points, t is the *t*-value at the 95% probability level, cv is the coefficient of variation in DBH, AE is the allowable sampling error in DBH at point sampling (this research used 10%).

In each forest type, all the selected tree species were grouped into 10 groups based on their wood density. The species with the highest IVI in each group was selected as a representative of the group for tree data and wood sample collection. Each selected species was further classified into one of three diameter classes (small, medium, and large) (15 tree samples in each species, 12 tree samples for establishing the equation process, and tree samples for the validation process. The total was 450 sample trees, 360 sample trees for establishing the equation process and 90 sample trees for the validation process).

The bole of each sample tree was measured for the stem diameter by 2 m sections from the base to the first major branch to calculate the tree bole volume. The wood samples of the selected species with the highest IVI, as described above, were collected in the sample tree bole in the north and east directions using an increment borer or a handsaw at 1.3 m height (2 wood samples in each tree for a total of 900 wood samples) to determine the carbon fraction. The wood samples were collected only at 1.3 m height as the literature indicated that the carbon fraction did not vary significantly along the stems [24].

### 2.3. Sample Preparation and Carbon Fraction Analysis

Wood sample preparation: This process was for estimating the wood carbon fraction in sample trees. The wood wet volume of the collected sample wood was calculated using Newton's formula Equation (2) [21,25]:

$$V_t = \sum_{i=1}^{n} \frac{L}{6} (Ab_i + 4Am_i + Au_i) \tag{2}$$

where, $V_t$ is the tree wet volume, $Ab_i$ is the cross-sectional area at the base of stem segment i, $Am_i$ is the cross-sectional area at the middle of stem segment i, $Au_i$ is the cross-sectional area at the upper of stem segment i, and L is the length of stem segment i (m).

The wet volumes of the wood samples were calculated. The wood samples were weighed and dried in an air-dry oven at 80 °C for 24–48 h until their weights became constant to determine the final dry weights. Two dry samples from the same tree were then pulverized together using a crushing machine to obtain a 100 g dry-weight sample.

Carbon fraction analysis: A 100 g pulverized sample was analyzed for carbon fraction using combustion methods via a carbon analyzer (e.g., PerkinElmer 2400 series II CHNS/O Elemental Analyzer), recommended by Kraenzel et al. and Wulzler et al. [26,27].

*2.4. Data Analysis*

(1) Carbon storage in the wood samples: This process was calculated using the relationships between carbon fraction and wood sample dry weight.

The carbon proportion (carbon fraction) was obtained as a percentage of dry weight using the method described by Duangsathaporn et al. [14] and Khantawan et al. [28] to convert the carbon fraction to carbon weight in a wood sample Equation (3):

$$C_c = C_w \times W_d \tag{3}$$

where, $C_c$ is the weight of carbon in a wood sample core (kg), $W_d$ is the dry weight of a wood sample core (kg), $C_w$ is the carbon fraction in a sample core (%).

Furthermore, the carbon wood sample and carbon fraction in each species was used to estimate the carbon stored in the standing tree using Equation (4):

$$C_t = \frac{C_c}{V_w} \times V_t \tag{4}$$

where, $C_t$ is the weight of carbon in a standing sample tree bole (kg), $C_c$ is the weight of carbon in a wood sample core (kg), $V_w$ is the wet volume of the wood sample core, and $V_t$ is the wet volume of the standing tree bole.

(2) The calculation of standing tree carbon stock: This process calculated the standing tree carbon storage in tree samples to estimate the carbon equation. The aboveground standing tree carbon was determined through three steps.

In the first step, a piece of sample tree in tree bole was used to estimate the bole volume and carbon. The wet bole volume (V) of every sample from a total of 362 sample trees was calculated using Smalian's formula Equation (5) [21,25], and the carbon stock in each wood sample core was then estimated using the dry weight carbon in the wood sample core multiplied by the carbon fraction in each wood sample core [14]. The whole-bole carbon stock of each sample tree was then calculated using the proportion of dry weight carbon in a wood sample core and the wet volume of the wood sample core multiplied by the wet volume of the standing tree bole Equation (4).

$$V_t = \sum_{i=1}^{n} \frac{L}{2}(Ab_i + Au_i) \tag{5}$$

where, $Ab_i$ is the cross-sectional area at the base of the stem segment i, $Au_i$ is the cross-sectional area at the upper of the stem segment i, and L is the length of the stem segment I (m).

In the second step, the branch and leaf carbon stock were estimated using the leaf and branch biomass of the tree, estimated using the standard equation multiplied by the carbon fraction. The equation recommended by Tsutsumi et al. was used to estimate the branch and leaf biomass for trees from the DEF [29], and the equation recommended by Ogawa et al. was used to estimate the branch and leaf biomass for trees from the MDF and DDF [30]. These equations for estimating carbon stock in leaves and branches in each tree are shown in Table 1.

In the third step, the aboveground carbon stock in each sample tree was obtained by combining stem, branch, and leaf carbon stock. This was then used to develop the tree carbon storage equations.

(3) The construction of standing tree carbon equations: The equations to estimate the aboveground standing tree carbon were constructed using the model $C = aDBH^bH^c$, where C is the standing tree carbon stock, DBH is the diameter at breast height, H is the total height, and a, b and c are model parameters to be estimated using Minitab statistics program [31]. The model parameters were estimated using log transformation and linear multiple regression.

**Table 1.** The equation for estimating carbon storage in leaves and branches in individual trees.

| Forest Type | Equation | Location | Source |
|---|---|---|---|
| Dry evergreen forest | $W_b = 0.00893(DBH^2H)^{0.977}$<br>$W_l = 0.0140(DBH^2H)^{0.669}$ | Phitsanulok Thailand | Tsutsumi et al., 1983 [29] |
| Dry deciduous forest | $W_s = 0.0396(DBH^2H)^{0.9326}$<br>$W_b = 0.003487(DBH^2H)^{1.0270}$<br>$W_l = (28.0/W_{sb} + 0.025)^{-1}$ | Nakhon Ratchasima Thailand | Ogawa et al., 1965 [30] |
| Mixed deciduous forest | $W_s = 0.0396(DBH^2H)^{0.9326}$<br>$W_b = 0.003487(DBH^2H)^{1.0270}$<br>$W_l = (28.0/W_{sb} + 0.025)^{-1}$ | Nakhon Ratchasima Thailand | Ogawa et al., 1965 [30] |

**Remark**: $W_s$ is the biomass of the stem (kg/tree), $W_b$ is the biomass of the branches (kg/tree), $W_l$ biomass of leaves (kg/tree), $W_{sb}$ is the biomass of the stem + biomass of branches (kg/tree), DBH is the diameter at breath height, and H is the height of the tree.

Standing trees data were divided into 2 groups, 80% for established standing trees equations and 20% for validation of the study equations using a *t*-test statistical analysis. The equations were fitted for each forest type. In order to select the optimal tree carbon equations, statistics which included the coefficient of determination ($R^2$), standard error of estimate (SE), F-value, and significance value (*p*-value, $\alpha \leq 0.05$), were evaluated. The normality of the model residuals was also examined using a basic program of statistics. The validation technique [32] was used to verify the accuracy of the equation by calculating the carbon storage in 20% of the collected samples. The methodology used to calculate the carbon stock of standing trees was also applied as described in the subsection on the calculation of standing tree carbon stock within the data analysis section. The carbon storage in each method was compared with the carbon from the established equation using the *t*-test statistical analysis. The equations of the three forest types were compared with the general equation for all species/wood density groups. This was done by calculating the relative differences and statistics between the mean of the equations of the three forest types and the optimal forest type equations. Data from 30 randomly selected sample trees were used to test the differences between the optimal equation and forest-type equations and compared the previous equation and present equation using the *t*-test analysis.

(4) Estimated stand carbon stock: All trees in the point sampling inventory from Section 2.1 were used for calculating the stand carbon storage. The carbon stock per hectare (ha) at each sampling point was estimated by summing the estimated carbon content of the sample tree and expressing it on a per unit area basis for the major forest types in the study area, using the Equations (6)–(9) adapted from van Laar and Akça [20].

$$C_p = BAF \times \sum_{i=1}^{n} \frac{C_i}{BA_i} \tag{6}$$

where, $C_p$ is the carbon stock at the sampling point (kg/ha), BAF is the basal area factor, $C_i$ is the carbon storage in tree i of point sampling, and $BA_i$ is the basal area in tree i of point sampling.

$$C_a = A_t \times \overline{C} \tag{7}$$

where, $C_a$ is the mean carbon stock in forest area, $A_t$ is the forest area in the study, and $\overline{C}$ is the average carbon stock in all sampling points.

$$C_t = C_a \pm t \times SE_{C_a} \tag{8}$$

and

$$SE_{C_a} = A_t \times SE_{\overline{c}} \tag{9}$$

where, $C_t$ is the carbon stock of the forest area, $SE_{c_a}$ is the standard error in the stand carbon stock of the forest area, and $SE_{\bar{c}}$ is the standard error of the mean carbon stock in the forest area.

## 3. Results

### 3.1. Forest Inventory for Building a Species List, Sample Trees Selection, and Wood Sample Extraction

A 54-sample fixed grid of size 3 × 3 km was established in the Mae Huad sector. Forty-four sampling points fell in the forested area and were classified under either of the three forest types, while the remaining 10 sampling points were in the agriculture field. Seventy-six tree species were found in the Mae Huad sector, with 46 tree species in the MDF, 18 in the DDF, and 31 in the DEF. The IVI was calculated and used to classify and select the sample trees. The highest species IVI in the MDF were *Xylia xlocarpa*, *Tectona grandis*, and *Prerocapus macrocapus*. In the DDF, they were *Shorea siamensis*, *Shorea obtusa*, and *P. macrocarpus*, and in the DEF, they were *Mallorus macrostachyus*, *Hopea odorata*, and *Duabanga grandiflora*. The *X. xylocarpa*, *Dalbergia oliveri*, *P. macrocarpus*, *Terminalia corticosa*, *Terminalia. alata*, and *Quercus kerri* were found in all three forest types. Within the three forest types and 10 species groups for each forest type, the sample trees were grouped into wood density classes for a total of 30 groups. In each group, the selected species had the highest value of IVI for the sample tree species. The 10 sample tree species per forest type were classified into three DBH classes (small, medium, and large) from inventory data covering a DBH range from 4.50 to 147.00 cm. The wood density range and representative tree species are shown in Table 2.

**Table 2.** Number of sample trees, carbon content, and carbon stock in the sample trees.

| Forest Type | Density Class | Wood Density Range * (kg/m³) | Representative Species (Scientific Name) | DBH Range (cm) | No. Sample trees | %Carbon | Carbon Stock in Sample Trees (Stem + Branch + Leaf) (kg) | | |
|---|---|---|---|---|---|---|---|---|---|
| | | | | | | | min | max | Average |
| Mixed deciduous forest | 1 | 283–385 | *Cananga latifolia.* | 4.50–43.00 | 15 | 47.75 | 23.90 | 454.29 | 163.37 |
| | 2 | 386–488 | *Litsea glutinosa* | 4.50–62.40 | 15 | 46.86 | 38.65 | 1297.63 | 505.81 |
| | 3 | 489–591 | *Lannea coromandelica* | 4.50–58.00 | 16 | 45.75 | 11.19 | 1252.27 | 407.75 |
| | 4 | 592–694 | *Tectona grandis* | 4.50–71.00 | 16 | 49.66 | 8.18 | 1385.72 | 589.75 |
| | 5 | 695–797 | *Albizia odoratissima* | 4.50–42.50 | 15 | 46.84 | 12.17 | 502.36 | 186.04 |
| | 6 | 798–900 | *Terminalia nigrovenulosa* | 4.50–61.29 | 16 | 47.13 | 38.10 | 1161.93 | 402.95 |
| | 7 | 901–1003 | *Pterocarpus macrocarpus* | 4.50–61.50 | 15 | 48.41 | 20.22 | 1489.53 | 445.37 |
| | 8 | 1004–1106 | *Xylia xylocarpa* | 4.50–66.80 | 15 | 48.03 | 29.40 | 1340.78 | 489.37 |
| | 9 | 1107–1209 | *Dalbergia oliveri* | 4.50–42.80 | 17 | 47.13 | 14.89 | 724.67 | 264.32 |
| | 10 | 1210–1312 | *Terminalia corticosa* | 4.50–66.30 | 15 | 48.55 | 22.87 | 2006.99 | 590.11 |
| Dry dipterocarp forest | 1 | 401–485 | *Mitragyna brunonis* | 4.50–41.00 | 15 | 47.57 | 16.05 | 496.35 | 189.90 |
| | 2 | 486–570 | *Bridelia pierrei* | 4.50–25.80 | 12 | 47.16 | 6.12 | 186.67 | 70.35 |
| | 3 | 571–655 | *Gardenia sootepensis* | 4.50–32.40 | 15 | 46.06 | 23.53 | 680.13 | 175.37 |
| | 4 | 656–740 | *Haldina cordifolia* | 4.50–41.9 | 15 | 48.26 | 8.86 | 604.76 | 177.12 |
| | 5 | 741–825 | *Dipterocarpus obtusifolius* | 4.50–42.50 | 15 | 47.62 | 9.77 | 505.84 | 145.02 |
| | 6 | 826–910 | NA (not applicable) | | | | | | |
| | 7 | 911–995 | *Pterocarpus macrocarpus* | 4.50–61.50 | 15 | 48.41 | 20.22 | 1489.53 | 445.37 |
| | 8 | 996–1080 | *Shorea siamensis* | 4.50–58.20 | 15 | 46.76 | 11.99 | 1148.05 | 438.29 |
| | 9 | 1081–1165 | *Dalbergia oliveri* | 4.50–42.80 | 17 | 47.13 | 14.89 | 724.67 | 264.33 |
| | 10 | 1166–1250 | *Terminalia corticosa* | 4.50–66.30 | 15 | 48.55 | 23.87 | 2006.99 | 590.11 |
| Dry evergreen forest | 1 | 388–474 | *Duabanga grandiflora* | 4.50–147.00 | 15 | 46.92 | 59.58 | 8673.15 | 2572.99 |
| | 2 | 475–561 | *Croton roxburghii* | 4.50–42.00 | 15 | 47.77 | 16.81 | 353.85 | 130.37 |
| | 3 | 562–648 | *Careya sphaerica* | 4.50–38.30 | 15 | 47.47 | 12.97 | 256.03 | 117.36 |
| | 4 | 649–735 | *Artocarpus lakoocha* | 4.50–47.30 | 15 | 48.31 | 13.12 | 915.60 | 269.66 |
| | 5 | 736–822 | *Cratoxylum formosum* | 4.50–22.60 | 15 | 46.83 | 8.45 | 119.54 | 42.31 |
| | 6 | 823–909 | *Anogeissus acuminata* | 4.50–71.70 | 15 | 46.81 | 56.61 | 2365.23 | 761.73 |
| | 7 | 910–996 | *Pterocarpus macrocarpus* | 4.50–61.50 | 15 | 48.41 | 20.22 | 1489.53 | 445.37 |
| | 8 | 997–1083 | *Terminalia alata* | 4.50–50.00 | 15 | 45.75 | 25.75 | 1271.14 | 410.04 |
| | 9 | 1084–1170 | *Xylia xylocarpa* | 4.50–66.80 | 15 | 48.03 | 29.40 | 1340.78 | 489.37 |
| | 10 | 1171–1257 | *Quercus kerrii* | 4.50–43.7 | 15 | 45.43 | 11.78 | 499.12 | 181.19 |

Source: * Forest Research and Development Bureau (2007; 2010) [33,34].

### 3.2. Wood Carbon Fraction Analysis

The carbon fraction of trees in the NDF analyzed using combustion techniques from the PerkinElmer 2400 series II CHNS/O Elemental Analyzer in the laboratory was between 45.75% and 49.66%, with an average of 47.43%. The carbon fraction of MDF for 10 species and 155 samples ranged from 45.75% to 49.66%, with an average of 47.61%. The third highest carbon fraction in each MDF sample species was *Tectona grandis*, 49.66%; *Terminalia Corticosa*, 48.55%; and (3) *Lannea coromandelica*, 45.75%, respectively. The carbon fraction of DDF for 9 species and 134 samples ranged from 46.06% to 48.55% with an average of 47.50%, and each third highest species carbon fraction was *T. corticosa*, 48.55%, *Pterocarpus macrocarpus*, 48.41%, and *Haldina cordifolia*, 48.26%, respectively. The carbon fraction of DEF for 10 species and 150 samples ranged from 45.43% to 48.41% with an average of 47.17%, and each third highest species carbon fraction was *P. macrocarpus*, 48.41%, *Artocarpus lakoocha*, 48.31%, and *Xylia xylocarpa*, 48.03% respectively. The carbon fraction of sample trees in the Mae Huad sector, NDF, are listed in Table 2.

### 3.3. Carbon Storage

The standing trees' carbon stock in each species was calculated using Equations (3)–(5), and the branch and leaf carbon was calculated using the equations in Table 1. Aboveground carbon stock in sample trees of NDF ranged between 6.12 and 8673.15 kg. In MDF, the aboveground carbon stock from 155 sample trees ranged from 8.18 to 2006.99 kg. The aboveground carbon stock in DDF from 134 sample trees ranged from 6.12 to 2006.99 kg, and the aboveground carbon stock in DEF from 150 sample trees ranged from 8.45 to 8673.15 kg. The carbon stock by tree component and the aboveground carbon stock per cubic meter are shown in Table 2. All aboveground carbon stock sample tree data were used to develop the standing tree carbon stock equations using regression analysis (Section 3.4).

### 3.4. Standing Tree Aboveground Carbon Equations

The aboveground carbon stock sample data were used to establish standing tree carbon equations for MDF, DDF, and DEF and a general equation for all the species/wood density groups. Eighty percent of sample standing trees data were used to estimate equations. Multiple regression analysis was used, where the dependent variable was carbon stock (C), and the independent variables included DBH and H. The two variables, DBH and H, both showed high relationships to C more than only DBH or only H (Table 3).

**Table 3.** The regression equations tested to estimate the standing tree carbon stock.

| Forest Type | Variables | Statistic Criterion Value | | | | Remark |
|---|---|---|---|---|---|---|
| | | $R^2$(%) | SE | F-Value | *p*-Value | |
| MDF | DBH | 0.9604 | 0.11 | 1708.54 | <0.001 | |
| | H | 0.7361 | 0.29 | 426.8 | <0.001 | |
| | DBH, H | 0.9699 | 0.10 | 1963.05 | <0.001 | * Best variables |
| DDF | DBH | 0.9259 | 0.16 | 1649.54 | <0.001 | |
| | TH | 0.7303 | 0.30 | 357.5 | <0.001 | |
| | DBH, H | 0.9405 | 0.14 | 846.14 | <0.001 | * Best variables |
| DEF | DBH | 0.9633 | 0.13 | 3888.38 | <0.001 | |
| | TH | 0.7039 | 0.36 | 351.89 | <0.001 | |
| | DBH, H | 0.9770 | 0.10 | 2482.67 | <0.001 | * Best variables |
| NDF | DBH | 0.9526 | 0.13 | 4779.48 | <0.001 | |
| | TH | 0.7171 | 0.32 | 1107.98 | <0.001 | |
| | DBH, H | 0.9611 | 0.12 | 3544.38 | <0.001 | * Best variables |

* Shown the best variable for constructed the equation.

The equations for the MDF, DDF, and DEF were estimated from 10, 9, and 10 species, respectively, and the general equation was estimated from 24 species. The DBH range of the trees used in the construction of equations for MDF, DDF, DEF, as well as the general equation, was between 8.70 and 71.00 cm, 10.00 and 66.80 cm, 9.70 and 147.00 cm, and 8.70 and 147.00 cm, respectively. The suitable equations are shown in Table 4.

**Table 4.** Carbon stock equations, DBH range, and statistical goodness of fit values of the constructed general equation.

| Forest Type | Equation | DBH Range (cm) | $R^2$ (%) | SE | F-Value | *p*-Value |
|---|---|---|---|---|---|---|
| MDF | $C_{MDF} = 0.0194 DBH^{2.2152} H^{0.5580}$ | 8.70–71.00 | 0.9699 | 0.10 | 1963.05 | <0.01 |
| DDF | $C_{DDF} = 0.0132 DBH^{2.1570} H^{0.7630}$ | 10.00–66.80 | 0.9405 | 0.14 | 846.148 | <0.01 |
| DEF | $C_{DEF} = 0.0185 DBH^{2.1371} H^{0.6804}$ | 9.70–147.00 | 0.9770 | 0.10 | 2482.67 | <0.01 |
| NDF | $C_{NDF} = 0.017754 DBH^{2.1899} H^{0.6260}$ | 8.70–147.00 | 0.9611 | 0.12 | 3544.38 | <0.01 |

**Remark** $C_{NDF}$, $C_{MDF}$, $C_{DDF}$, and $C_{DEF}$ indicate the aboveground standing tree carbon stock in the general equation for all species/wood density groups in the NDF, MDF, DDF, and DEF, respectively (kg/tree), while DBH is the diameter at breast height of the tree (cm), H is the total height of the tree (m).

The coefficient of determination ($R^2$), standard error of estimate (SE), F-value, and significant value (*p*-value, $\alpha \leq 0.05$) to determine the best fit equations were determined for each forest type. The $R^2$ values for the equations constructed for MDF, DDF, DEF, and NDF were 0.9678, 0.9412, 0.9770, and 0.9642, while the SE was 0.101, 0.139, 0.100, and 0.114, respectively. The F-value was 2281.89, 1048.28, 2127.52, and 5870.00, respectively, while the *p*-value for all the equations was less than 0.01, which was highly significant. All the related statistical values for each forest type are shown in Table 4.

The residuals between the actual and estimated carbon stock for the various values of carbon stock are shown in Figure 3. Residuals for the overall model can be seen to be unbiased, as were for all species in the MDF, DDF, DEF, and all species in the NDF. In other words, the errors are distributed uniformly with no apparent dependence on any of the potential predictors.

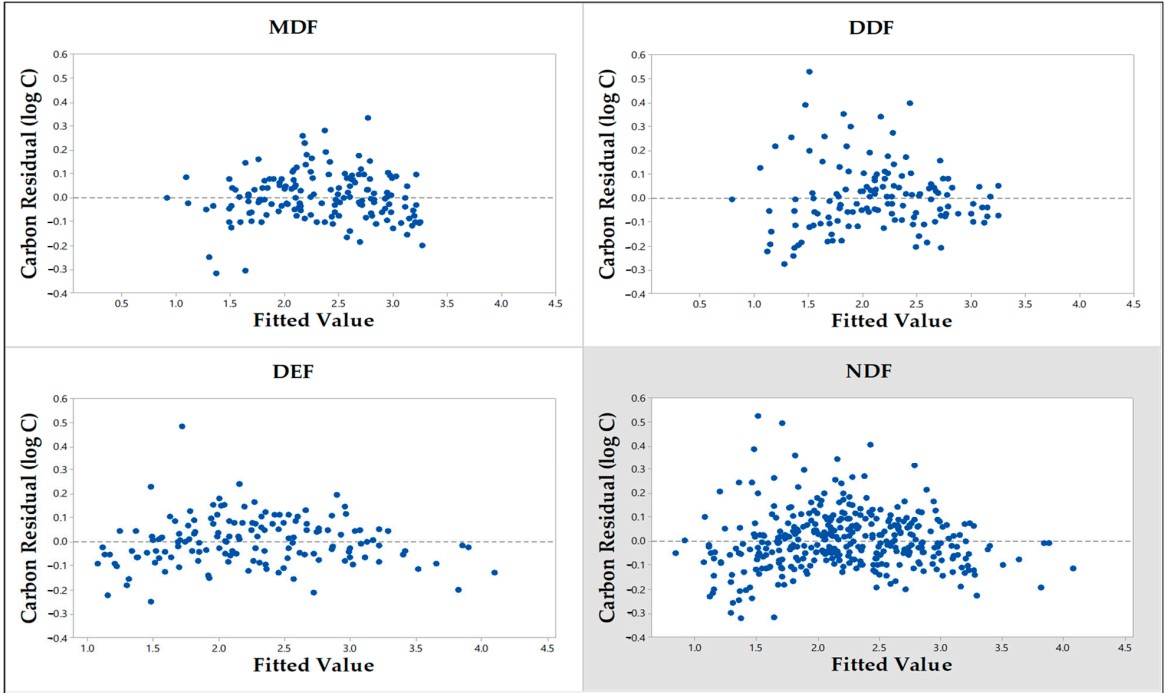

**Figure 3.** Residuals or the difference between observed and predicted aboveground carbon content of the selected trees: MDF, DDF, DEF, and Mae Huad sector NDF.

Sample trees were used to estimate the tree carbon stock for each forest type using tree DBH and H. The carbon storage of tree samples was estimated using the equations constructed for the MDF, DDF, DEF, and the optimal equation of this project. The equations to estimate the carbon content in the MDF, DDF, and DEF were similar to the optimal equation. The relative difference between the two carbon equations was between 0.088 and 2.416%, 0.050 and 2.545%, and 1.076 and 2.191%, respectively. The validity of the constructed equations was confirmed by employing a *t*-test statistical analysis to compare the carbon storage values of 30 trees, as estimated by the constructed equations and the carbon estimations explained in Section 3.3 (Figure 4). The results of the *t*-test revealed that there was no significant difference in carbon storage between these two groups (Table 5). The comparison of the mean between the equations of the three forest types and the optimal equation was not a significant difference.

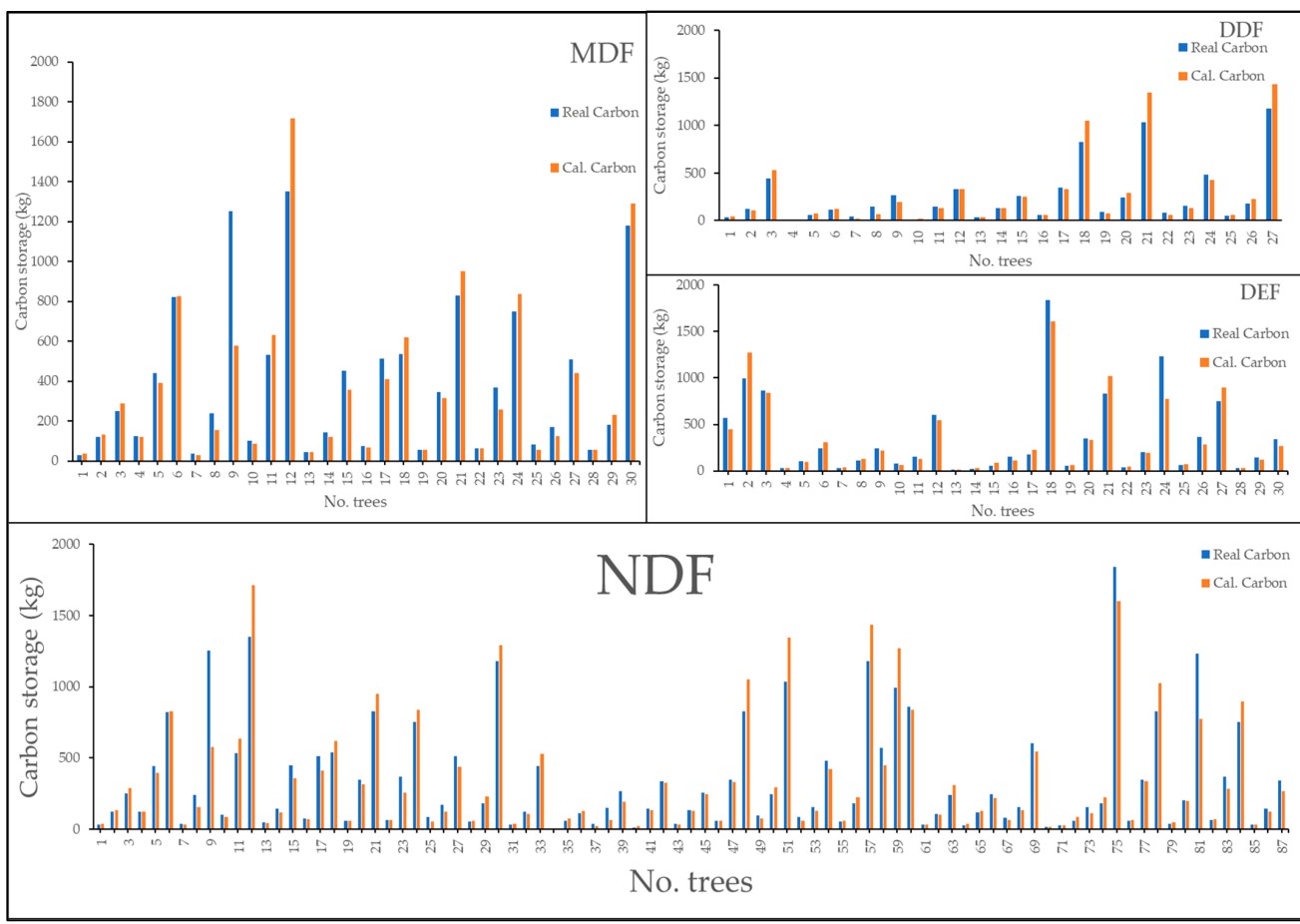

**Figure 4.** The comparison of the standing tree carbon storage. Real Carbon is actual carbon as estimated in Section 3.3, and Cal. Carbon is carbon storage estimated with the constructed equations.

*3.5. Forest Stand Carbon Stock in the NDF*

The standing tree carbon stock was used to calculate the carbon stock in the NDF forest stand using the equation from van Laar and Akça [20]. The carbon stock of the 44 sampling points ranged from 6.15 to 175.64 tons/ha, with an average per point of 61,837.96 kg/ha (Table 6). Carbon stock per hectare by forest type in the DDF, MDF, and DEF was 142 tons/ha, 53.02 tons/ha, and 12 tons/ha, respectively. The carbon stock in the MDF was approximately between 0.932 and 1.4 million tons (1.20 mean million tons), and that in the DDF was approximately between 0.289 and 0.454 million tons (0.371 mean million tons), and that in the DEF was approximately between 0.049 and 0.078 million tons (0.063 mean million tons). The carbon storage in the Mae Huad sector forest was approximately between 1.27 and 1.99 million tons (average of 1.632 million tons).

**Table 5.** Value of *t*-test between the verification and the validation value.

| Forest Type | Statistic | Variable 1 | Variable 2 |
|---|---|---|---|
| MDF | Mean | 388.49 | 376.33 |
| | Observations | 30.00 | 30.00 |
| | P(T ≤ t) two-tail | 0.67 | |
| DDF | Mean | 254.50 | 278.85 |
| | Observations | 27.00 | 27.00 |
| | P(T ≤ t) two-tail | 0.18 | |
| DEF | Mean | 464.87 | 529.91 |
| | Observations | 30.00 | 30.00 |
| | P(T ≤ t) two-tail | 0.43 | |
| NDF | Mean | 373.25 | 399.04 |
| | Observations | 87.00 | 87.00 |
| | P(T ≤ t) two-tail | 0.39 | |

**Table 6.** Carbon stock (CS) per sample point in the Mae Huad sector.

| Point NO. | CS (kg/ha) | Forest Type | Point NO. | CS (kg/ha) | Forest Type | Point NO. | CS (kg/ha) | Forest Type |
|---|---|---|---|---|---|---|---|---|
| 1 | 88,287.97 | MDF | 16 | 126,052.64 | DDF | 31 | 9129.03 | DEF |
| 2 | 17,097.92 | DEF | 17 | 20,736.35 | MDF | 32 | 23,084.19 | MDF |
| 3 | 69,643.47 | MDF | 18 | 42,646.39 | MDF | 33 | 64,055.07 | MDF |
| 4 | 7201.58 | DEF | 19 | 47,857.28 | MDF | 34 | 22,489.32 | DEF |
| 5 | 128,574.47 | DDF | 20 | 165,600.93 | DDF | 35 | 106,388.19 | MDF |
| 6 | 100,360.65 | DDF | 21 | 38,493.44 | MDF | 36 | 67,442.07 | MDF |
| 7 | 58,061.08 | MDF | 22 | 11,156.50 | MDF | 37 | 148,058.31 | DDF |
| 8 | 6152.98 | DEF | 23 | 59,036.21 | MDF | 38 | 43,768.32 | MDF |
| 9 | 27,286.41 | MDF | 24 | 162,671.46 | DDF | 39 | 79,015.26 | MDF |
| 10 | 63,391.82 | MDF | 25 | 38,514.85 | MDF | 40 | 70,904.44 | MDF |
| 11 | 59,217.74 | MDF | 26 | 42,606.07 | MDF | 41 | 133,507.34 | DDF |
| 12 | 46,099.49 | MDF | 27 | 63,438.04 | MDF | 42 | 57,309.07 | MDF |
| 13 | 28,610.93 | MDF | 28 | 63,581.88 | MDF | 43 | 46,216.42 | MDF |
| 14 | 8623.92 | DEF | 29 | 9828.38 | DEF | 44 | 11,595.44 | DEF |
| 15 | 61,435.68 | MDF | 30 | 175,641.25 | DDF | Average | 61,837.96 kg/ha | |

## 4. Discussion

### 4.1. Carbon Fraction

Normally, the carbon fraction is assumed to be 50% of a tree's total biomass [6,35,36]. The carbon fraction in this study ranged from 45.75% to 49.66%, with an average of 47.43%. This is less than the normally assumed value but more than the Intergovernmental Panel on Climate Change (IPCC) carbon fraction value of 47% of tree biomass [5]. However, much research has explained that the variations in carbon fraction estimates might result from the methods used for different species, the components of a tree or a stand used (stem, roots, and leaves), and the age of the stand [6,37]. For example, the study by Thomas and Martin, which reports on the difference in carbon fraction in parts of trees, shows 37%, 76%, 48%, 81%, and 63%, respectively, of the variation in bark, branch, twig, coarse root, and fine root carbon fraction values [6]. IPCC reports confirmed the difference in carbon fractions. The amounts of components of wood tissues such as cellulose, hemicellulose, lignin, and a variety of nonstructural chemicals resulted in different amounts of carbon by mass [38]. The carbon fraction of trees growing in plantations in Thailand was reported by Diloksumpun and Staporn, who estimated the carbon stock through combustion techniques. They found

that the carbon fraction for *eucalyptus* spp. was 48.36% [39], while Duangsathaporn et al., who estimated the sequestered carbon in standing teak trees in the Thong Phaphum teak plantation through combustion techniques, reported a carbon fraction of 46.58% [14].

We investigated the relationship between average carbon per cubic meter and wood density; higher-density wood had a higher carbon content per cubic meter (Table 7). The relationship between average carbon weight per wood volume (kg/m$^3$) and wood density class was analyzed using a linear relationship. This linear relationship was not significant (Figure 5), but there was a trend showing variation in the wood samples due to different wood elements (e.g., lignin, cellulose, and hemicellulose) in a unit of tree sample volume. Thus, the carbon stock per volume will be different, similar to the study of Campbell and Sederoff, who found differences in lignin in different tree species in sample trees [40]. A study by Navarro et al. also found that indirect indicators of wood density and carbon fraction affected carbon storage, as high wood density in some species of tropical forests was shown to have high carbon content [41]. Other studies have also not found such a relationship between carbon storage and wood density. For example, Weber et al. studied the variations between tree growth, density, and carbon concentration and did not find any significant relation between wood density and carbon concentration [42].

**Table 7.** The average carbon per cubic meter in sample trees and wood density.

| Density Class | Carbon in a Cubic Meter (kg) in MDF | | | | Carbon in a Cubic Meter (kg) in DDF | | | | Carbon in a Cubic Meter (kg) in DEF | | | |
|---|---|---|---|---|---|---|---|---|---|---|---|---|
| | Stem | Branch | Leaf | Total | Stem | Branch | Leaf | Total | Stem | Branch | Leaf | Total |
| 1 | 220.34 | 60.06 | 17.76 | 298.15 | 289.33 | 73.00 | 27.95 | 390.27 | 227.50 | 167.51 | 9.03 | 404.03 |
| 2 | 281.49 | 78.52 | 22.89 | 382.90 | 284.83 | 84.54 | 33.00 | 402.37 | 260.01 | 164.63 | 10.81 | 435.45 |
| 3 | 221.62 | 48.75 | 14.48 | 284.85 | 365.20 | 89.50 | 34.70 | 489.40 | 257.70 | 187.24 | 12.10 | 457.04 |
| 4 | 293.32 | 94.68 | 27.66 | 415.66 | 301.62 | 105.00 | 39.93 | 446.55 | 314.15 | 187.81 | 11.52 | 513.47 |
| 5 | 319.23 | 68.76 | 20.35 | 408.33 | 317.04 | 79.18 | 30.63 | 426.85 | 314.15 | 223.06 | 15.37 | 552.57 |
| 6 | 354.41 | 108.68 | 31.72 | 494.81 | NA (not applicable) | | | | 307.47 | 195.46 | 10.48 | 513.40 |
| 7 | 385.07 | 98.06 | 28.83 | 511.96 | 385.07 | 98.06 | 28.83 | 511.96 | 385.07 | 98.06 | 28.83 | 511.96 |
| 8 | 345.14 | 84.59 | 24.77 | 454.51 | 350.46 | 79.55 | 29.77 | 459.77 | 348.71 | 162.78 | 9.69 | 521.17 |
| 9 | 386.94 | 68.10 | 20.10 | 475.13 | 386.94 | 68.10 | 20.10 | 475.13 | 345.14 | 84.59 | 24.77 | 454.51 |
| 10 | 314.15 | 84.83 | 24.92 | 423.91 | 314.15 | 84.83 | 24.92 | 423.91 | 342.86 | 128.51 | 9.03 | 480.40 |

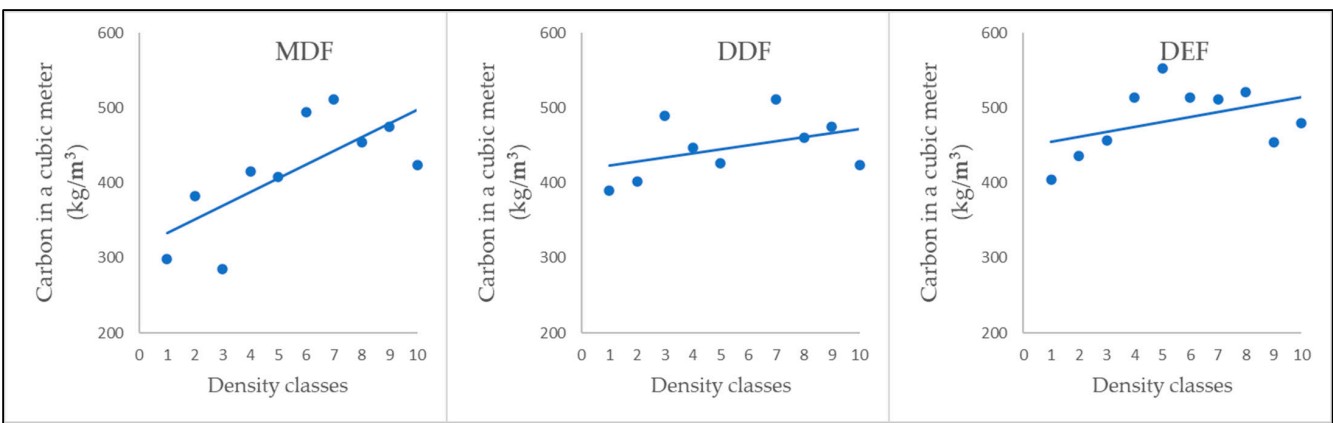

**Figure 5.** The relationships shown between stored carbon per cubic meter and wood density class.

### 4.2. Aboveground Carbon Equation

Generally, equations for estimating carbon storage use indirect methods, such as the product of tree biomass and carbon fraction, to evaluate the carbon stored in a standing tree. In Thailand, to estimate carbon storage, Viriyabuncha used an aggregate of six equa-

tions in a mixed-species forest [12], Yuan et al. used 11 equations in a mixed-species forest [43], and the Department of Environmental Quality Promotion used six equations from trees in a mixed deciduous, dry dipterocarp, dry evergreen, pine, and mangrove forest which included mixed species [44]. All of these studies conducted in Thailand used destructive sampling methods [43]. Destructive methods involve the cutting down of all the trees in the sampling area and measuring the weight of the different components, such as the tree trunk, leaves, and branches, and measuring the oven-dried weight of these components [45,46]. The carbon storage from the existing equation was similar to the new equations.

In this research, the equations for estimating carbon storage were constructed by combining the non-destructive method of Montès et al. [47] and the collection of the wood samples as proposed by Duongsathaporn et al. [14] and Khantawan et al. [28]. The regression technique was used to estimate the tree carbon storage in only the mixed deciduous, dry dipterocarp, and dry evergreen forest types. In addition, these equations can only be used for estimating the carbon stored in the standing trees with a DBH from 8.70 to 147.00 cm in the three tropical forest types. Thus, similar studies need to be done in other forest types in Thailand and with trees of a wider range of DBH. The advantage of the non-destructive sampling method is that, in Thailand, the destructive sampling method cannot be used as there has been an ongoing logging ban in natural forests since 1989 [48].

The most frequently used equations for estimating tree biomass in Thailand are those proposed by Ogawa et al. These equations were based on 90 standing sample trees in MDF and DDF (the coefficient of determination ($R^2$) = 0.9326) [30]. Another equation widely used is that of Tsutsumi et al., which used six standing sample trees in a DEF ($R^2$ = 0.97) [29]. All these equations are suitable for estimating tree biomass in the respective forest types, but the estimation of standing tree carbon is arduous as it still requires the estimation of the carbon fraction. Thus, the equations developed in this study are more suitable for directly estimating the carbon stored in standing trees in the MDF, DDF, and DEF forest types. The carbon equations were not developed by species as this would be too costly. There are many species in each forest type (46 tree species in the MDF, 18 in the DDF, and 31 in the DEF). Instead, the equations were developed using representative species in the various wood density classes.

## 5. Conclusions

The optimal aboveground carbon equations were formulated from a large sample of trees (155, 134, and 150 trees samples in MDF, DDF, and DEF, respectively) of various sizes and tree species (24 tree species in 3 forest types). We conclude that such equations can be used to estimate the carbon stocks in Thailand and in the assessment of carbon stock. However, the present carbon estimation did not cover other species in other forest types, such as mangrove forests. These are endeavors of future study.

## 6. Patents

This research was supported by the Asia-Pacific Network for Sustainable Forest Management (APFNet).

**Author Contributions:** Validation, Y.O. and P.P.; Data curation, K.P. and P.L.; Writing—original draft, N.S.; Writing—review & editing, K.D.; Supervision, K.D. All authors have read and agreed to the published version of the manuscript.

**Funding:** This research was funded by the Asia-Pacific Network for Sustainable Forest Management (APFNet) grant number [2015P6-THA-PD].

**Data Availability Statement:** Not applicable.

**Conflicts of Interest:** The authors declare no conflict of interest.

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
