# Peer review of "Formulating Equations for Estimating Forest Stand Carbon Stock for Various Tree Species Groups in Northern Thailand"

_forests, doi:10.3390/f14081584_

Round 1

Reviewer 1 Report

The manuscript titled “Formulating Equations for Estimating Forest Stand Carbon 2 Stock for Various Tree Species Groups at the Ngao Demonstration Forest, Northern Thailand” aimed to formulate the standing tree carbon equations to estimate the carbon stocks in three forest types based on four hundred thirty-nine wood samples. The authors have done a lot of work in collecting data, constructed carbon storage equations for three forests types, and the results are helpful to understand the changes in carbon storage. The paper uses carbon fraction to calculate the carbon stock of trees, which reduces the damage to trees and shows some innovation. However, there are some comments for the paper to be considered.

 Specific comments:

 1. The authors did a lot of work to carry out this research. The three carbon stock equations for three forests types (MDF, DDF and DEF) were built to calculate the carbon storage, and the general equation for all species in the NDF was also constructed. But I think the biggest problem of this article was that it did not take into account the differences between tree species. As we know, different species show differences in their growth processes and wood densities. Therefore, trees with the same DBH and height maybe do not have the equal carbon storages. So, I don’t think it’s accurate to construct the equations without considering the differences of the species. I suggest that the species could be the dummy variables to be added to the equations, which display the differences of species and improve the performance of the equations.

 2. In my understanding, wood sample cores were collected to calculate the carbon fractions at 1.3-meter height. Then, the weight of carbon in a standing sample tree bole was calculated based on the carbon fraction. However, the carbon fractions at different positions on trees are maybe not consistent. It’s questionable to use a fixed carbon fraction to calculate the carbon storage.

 3. The paper did not investigate the carbon stock of trees below ground, it is suggested to change "total carbon stock" to "aboveground carbon stock."

 4. The paper established the equations using all the measured data, but did not validate the equations. I suggest using leave-one-out cross-validation method to validate the equation.

 5. I think many references were too outdated, before 2020.

Author Response

Response to Reviewer 1 Comments

Point 1:The authors did a lot of work to carry out this research. The three carbon stock equations for three forests types (MDF, DDF and DEF) were built to calculate the carbon storage, and the general equation for all species in the NDF was also constructed. But I think the biggest problem of this article was that it did not take into account the differences between tree species. As we know, different species show differences in their growth processes and wood densities. Therefore, trees with the same DBH and height maybe do not have the equal carbon storages. So, I don’t think it’s accurate to construct the equations without considering the differences of the species. I suggest that the species could be the dummy variables to be added to the equations, which display the differences of species and improve the performance of the equations.

Response 1: There were many species in each forest type (18 to 46). Thus, the equations were not developed by species as this would be too costly. Instead, the equations were developed using species representative of various wood density classes. (Line 58)

Point 2: In my understanding, wood sample cores were collected to calculate the carbon fractions at 1.3-meter height. Then, the weight of carbon in a standing sample tree bole was calculated based on the carbon fraction. However, the carbon fractions at different positions on trees are maybe not consistent. It’s questionable to use a fixed carbon fraction to calculate the carbon storage.

Response 2: A reference showed no significant difference in carbon fraction along the stems. (Line 146-148 (Castaño-Santamaría, Javier, and Felipe Bravo. "Variation in Carbon Concentration and Basic Density Along Stems of Sessile Oak (Quercus Petraea (Matt.) Liebl.) and Pyrenean Oak (Quercus Pyrenaica Willd.) in the Cantabrian Range (Nw Spain)." Annals of Forest Science 69 (2012): 663-72.)

Point 3 The paper did not investigate the carbon stock of trees below ground, it is suggested to change "total carbon stock" to "aboveground carbon stock.

Response 3: We changed "total carbon" to "above-ground carbon".

Point 4 The paper established the equations using all the measured data, but did not validate the equations. I suggest using leave-one-out cross-validation method to validate the equation.

Response 4: We added the Validation in Methodology line 139, 212-214, and added the result in line 238-342.

Point 5 I think many references were too outdated, before 2020.

Response 5 We included references to outdated papers predating 2020, including prominent journals like Ogawa et al. 1965 and Tsutsumi et al. 1983, which were commonly utilized for estimating carbon storage in Thailand. Certain papers outlining specific research methodologies and functions remained unchanged, while others were changed.

Reviewer 2 Report

This article proposes an effective method to estimate carbon stock with high accuracy and build equations that can be used to estimate the carbon stocks in Thailand. However, several points need to be improved.

Major comments

Overall it is unclear to me what the novelty of this work is given that many studies have already investigated tree carbon stock, what exactly this study contributes to the literature that hasn’t been done before and the importance of this study should be stressed.

Introduction:

1. It seems that the first paragraph of the introduction is not so closely related to the content of the article, please revised it.

2. Summary of previous work is insufficient, and this manuscript lists defects of the existing works, and what targeted solutions are used to make up for these deficiencies are not clearly stated.

Materials and Methods

1. How many types of vegetation are there in the study area and why do you choose these three types? How about the other forest types? Since the fixed grid of 3×3 km covers the whole Mae Huad section, why not use all forest types in this region to obtain equations for more species?

2. When describing the study area, more details of natural forests are needed, such as temperature, precipitation, terrain, and soil since site conditions of vegetation are important for carbon Stock estimation. There is no need to specify the human-use land except for cropland.

4. It is good to see that this manuscript describes enough details of the method, and an overall flow chart is needed that would help readers understand the overall process.

Results

 1. The equations obtained based on this method are accurate, comparisons with other commonly used methods are suggested.

The quality of English needs to improving

Author Response

Response to Reviewer 2 Comments

Point 1: 1. Major comments Overall it is unclear to me what the novelty of this work is given that many studies have already investigated tree carbon stock, what exactly this study contributes to the literature that hasn’t been done before and the importance of this study should be stressed.

Response 1: This study addresses the weaknesses of the existing methods and equations used to estimate carbon stock in Thailand. This is mentioned in the last paragraph in the Discussion section.

Point 2: Introduction: (1). It seems that the first paragraph of the introduction is not so closely related to the content of the article, please revised it.

Response 2: The Introduction section has been revised. In particular, the sentences dealing with climate change have been deleted.

Point 3 . Summary of previous work is insufficient, and this manuscript lists defects of the existing works, and what targeted solutions are used to make up for these deficiencies are not clearly stated.

Response 3: The solutions to the defeects are summarized in the Discussion section.

Point 4 How many types of vegetation are there in the study area and why do you choose these three types? How about the other forest types? Since the fixed grid of 3×3 km covers the whole Mae Huad section, why not use all forest types in this region to obtain equations for more species?

Response 4: There were only 3 forest types in the study area as shown in the revised study area map Figure 2.

Point 5 When describing the study area, more details of natural forests are needed, such as temperature, precipitation, terrain, and soil since site conditions of vegetation are important for carbon Stock estimation. There is no need to specify the human-use land except for cropland.

Response 5 terrain, climate and soil descriptions of the study area have been added .

Point 6 It is good to see that this manuscript describes enough details of the method, and an overall flow chart is needed that would help readers understand the overall process.

Response 6 The methodology flowchart has been added as Figure 1.

Point 7 The equations obtained based on this method are accurate, comparisons with other commonly used methods are suggested.

Response 7 The new carbon equations were compared to the existing . The carbon contents from the existing equation were similar to the new equations.

Round 2

Reviewer 1 Report

1. I suggest that the authors should add some resently published references.

2. Please add some contents for the conclusion.

Author Response

Point 1: I suggest that the authors should add some recently published references.

Response 1: We added four published references (reference numbers 2, 3, 5 and 7) to the revised paper (Lines 426,427,429 and 433)

Point 2: Please add some contents for the conclusion.

Response 2: We added some remarks on the inventory results, carbon fraction values, sample trees and carbon equations validation in the Conclusions section. (Lines 405-410, and 413-414)
